# Utilising the Implementation of Integrated Care to Develop a Pragmatic Framework for the Sustained Uptake of Service Innovations (SUSI)

**DOI:** 10.3390/healthcare11121786

**Published:** 2023-06-16

**Authors:** Catherine Foley, Julaine Allan, Julia Lappin, Ryan Courtney, Sara Farnbach, Alexandra Henderson, Anthony Shakeshaft

**Affiliations:** 1National Drug and Alcohol Research Centre (NDARC), University of NSW (UNSW), Sydney, NSW 2052, Australia; 2Rural Health Research Institute, Charles Sturt University (CSU), Orange, NSW 2800, Australia; 3Department of Psychiatry and Mental Health, University of NSW (UNSW), Sydney, NSW 2052, Australia; 4Poche Centre for Indigenous Health, University of Queensland (UQ), Toowong, QLD 4066, Australia

**Keywords:** integrated care, translational, co-design, health services’ research, best-evidence models, implementation, psychiatric and addiction services

## Abstract

The provision of integrated care (IC) across alcohol and other drug (AOD) and mental health (MH) services represents the best practice, yet the consistent delivery of IC in routine practice rarely occurs. Our hypothesis is that there is no practical or feasible systems-change approach to guide staff, researchers, or consumers through the complex transition that is required for the sustained uptake of IC across diverse clinical settings. To address this gap, we combined clinical and consumer expertise with the best available research evidence to develop a framework to drive the uptake of IC. The goal was to develop a process that is both standardised by the best available evidence and can be tailored to the specific characteristics of different health services. The result is the framework for Sustained Uptake of Service Innovation (SUSI), which comprises six core components that are applied in a specified sequence and a range of flexible activities that staff can use to deliver the core components according to their circumstances and preferences. The SUSI is evidence-based and practical, and further testing is currently underway to ensure it is feasible to implement in different AOD and MH services.

## 1. Introduction

Integrated care (IC) is widely recognised as a best-practice service delivery approach for people who have multiple healthcare needs [1,2,3,4,5,6,7,8,9,10,11,12,13,14,15]. The definitions of IC describe treatment services working together, in contrast to definitions of integrated treatment (treatment for multiple conditions provided by one clinician or one service), parallel approaches (simultaneous treatments provided by separate services), and sequential approaches (completion of treatment in one service before a referral is made to another service) [16,17]. The aim of IC is to optimise treatment outcomes and healthcare experiences by identifying the spectrum of people’s needs at their first contact with a health service. Treatment plans can then be coordinated to meet those needs, rather than requiring people to independently access separate treatment options. The application of IC across mental health (MH) and alcohol and other drug (AOD) services, for example, is recommended in clinical practice guidelines from Australia [16,17], Canada [18], the US [19], and the UK [20] due to the prevalence rates for co-occurring disorders in specialist treatment settings reaching greater than 75%. For people accessing both MH and AOD services, the potential benefits of IC include a reduced relapse into substance abuse, fewer emergency hospital admissions, reduced deterioration of mental illness symptoms, and improved treatment experiences [21,22]. For MH and AOD services providing treatment, the key potential benefit is being able to provide high-quality treatment more efficiently by reducing treatment costs and clinical time and minimising the duplication of treatment across disparate services [2,4,6,8]. Despite the potential benefits of IC, however, its sustained uptake into routine practice remains challenging, and MH and AOD services typically operate independently of each other [1,2,3,17].

There are two possible reasons for the lack of sustained uptake of IC into routine practice. First, service providers may be reluctant to commit their scarce resources and training opportunities to IC because the empirical evidence supporting its effectiveness is still emerging [23], and the outcomes vary according to the specific combination of MH and substance use disorders [21,23,24,25,26]. For example, meta-analytic evidence shows that IC for alcohol misuse and depression achieves improved treatment outcomes relative to alcohol treatment alone but does not consistently improve the treatment outcomes for alcohol misuse and anxiety [27]. The limited published evidence on the effectiveness of IC for different combinations of co-occurring disorders does not preclude recommendation for its uptake, as endorsed in various published clinical guidelines [16,17,18,19,20], but it is a nuance in the evidence base that increases the difficulty in implementing IC into routine clinical practice. 

Second, it may be that the uptake of IC is not being guided by the application of a feasible, acceptable, and relevant framework for implementing change. Existing IC models exist, such as the Dual Diagnosis Clinician Shared Care Model [28], which demonstrates consumer preference for the treatment of both disorders at one time, and the Integrated Treatment for Co-Occurring Disorders Evidence-Based Practices (EBP) KIT [29], which does identify processes that help facilitate the uptake of IC, including the importance of incorporating practical organisational change. These models, however, focus on integrated treatment programs as opposed to service delivery programs, or they focus on the provision of care by one service and are not designed to facilitate the general adoption of IC into routine practice across diverse AOD and MH services. This means that, as yet, recognising the need for the improved treatment of co-occurring MH and substance use disorders, or the identification of IC as a best-practice treatment in national clinical guidelines and models of integration, has been sufficient to achieve meaningful and sustained change from independent care to IC in the routine delivery of clinical AOD and MH services. 

### Facilitating Change in Healthcare Services

While organisational change models, such as Lewin’s 3-Stage Model [30] and Kotter’s 8-Step Model [31], do identify the importance of planning for change in clinical settings, there is limited evidence to demonstrate how and why specific change models work [32,33]. This means those change models cannot sufficiently detail the type of activities that are most likely to create clinically meaningful and sustained change across multiple health service settings. A review of Kotter’s popular 8-Step Model of Change across fifteen years of literature, for example, established support for most of the steps individually [31]. What is lacking, however, is sufficient evidence demonstrating the validity of the steps collectively or evidence for the specified sequence in which the steps are applied [31]. As with the existing IC models, organisational change models are also largely focused on implementing change in one type of service [33], whilst qualitative studies of IC have shown that planning for change requires meaningful contributions from clinicians, consumers, and managers in multiple services at the same time [34]. Working simultaneously with MH and AOD services is critical for building a culture of collaboration across otherwise separate services, primarily because it increases the potential for staff at all levels of service provision to perceive change as empowering rather than disempowering and threatening [34]. Yet, few services have the time or resources to examine how to achieve genuine participation from clinicians, and the potential for innovation can be lost before it starts [35].

One approach to increasing the uptake of system-wide IC that shows promise across a range of healthcare services is the adaptation of evidence-based practice (EBP). This method involves the development of clinical care that is grounded in the best available research evidence whilst at the same time incorporating clinical and consumer expertise [36,37]. The more common adaptation of EBP targets individual patient care (personalized healthcare) and is often developed by individual clinicians and recorded in patients’ notes. The adaptation of EBP at a department or service level, however, aims to optimise the quality and consistency in healthcare across an entire service or sector. The latter is a process undertaken by a group of professionals rather than individual clinicians, and decision-making is informed by group discussion and consensus [38]. Additionally, this process requires the development of a protocol or framework and a detailed implementation plan to ensure the sustained implementation of agreed clinical changes at the service level [37,38]. As with the existing IC models and organisational change models, however, there is limited evidence on how the change transition required for EBP adaptations has been applied, nor is there adequate evidence about its effectiveness [37,38,39]. Specifying pragmatic details that give effect to the required change transition in a framework that services can both apply in a standardised way and tailor to their own circumstances is more likely to facilitate the uptake of sustained and measurable change across multiple services than simply describing the appropriate stages of change. 

Consequently, this study aimed to utilise the opportunity of implementing IC into routine clinical practice in MH and AOD services to articulate a co-designed, pragmatic framework for the Sustained Uptake of Service Innovations (SUSI): the SUSI framework.

## 2. Materials and Methods

### 2.1. Identifying the Core Components of the SUSI Framework

The development of the SUSI framework was informed by two key principles. First, the principle of evidence-based practice, which is defined as the combination of current best evidence with clinical and consumer expertise [36,37]. 

Current best evidence was determined by identifying the existing key principles of organisational change [30,31,32,33] and the existing features of IC models that are relevant to their implementation [16,17,18,19,20,21,22].Clinical and consumer expertise was determined in a Participatory Action Research (PAR) study conducted by the authors CF, AS, JA, JL, and RC, which identified the key issues for managers and clinicians in moving from the independent delivery of AOD and MH services to IC [34].

The second principle involved establishing a framework that can be tailored to different services and aligned with the adaptation of EBP at a service level. The authors previously developed treatment models comprising core components that are standardised by the best available research evidence and operationalised by activities that are identified by services and tailored to their own circumstances [40,41,42,43,44]. That approach was replicated for the development of the SUSI framework. Specifically, the data obtained by combining current literature with the findings of the PAR study were organised into core components and activities that the participating services identified as being important for the successful and sustained uptake of IC into practice and then articulated as the SUSI framework. The details of the PAR study [34] and literature review [45] are described elsewhere, the key results of which are summarised in Appendix A.

### 2.2. Establishing a Panel of Research and Clinical Experts to Construct the SUSI Framework

A research advisory group was formed during the PAR study [34] to provide evaluation support to the clinical research team and assist with ensuring the practice changes were aligned with the research evidence. The clinical research team invited appropriately skilled research staff from NSW Health, the National Drug and Alcohol Research Centre at UNSW, and the Centre for Rural and Remote Mental Health to join the group. The members included six invited representatives with research expertise, one director of AOD and MH services, and four members of the clinical research team who led the PAR study (two from MH and two from AOD). The members joined scheduled monthly teleconferences as often as was practical and/or communicated directly with the lead author (CF). Following the conclusion of the PAR study, the group led the development of the SUSI framework.

### 2.3. Organising the Core Components into a Sequential Framework

The SUSI framework was developed in alignment with the procedure for adapting EBP for groups into a framework [37,38,39]. Table 1 describes the activities undertaken. For example, the key elements of successful uptake were identified in Steps 1 and 2 (ask a searchable question and acquire information) and then reviewed and triangulated in Step 3 (appraise search results and draft a framework) and refined in Step 4 (disseminate and improve the framework, apply the evidence in practice).

The data appraised in Step 3 of the EBP procedure were triangulated [46] by:Mapping the key change transition processes that were identified in the PAR study (securing participation; developing relationships between teams; building ownership and confidence; and embedding sustainability mechanisms (at treatment and workforce levels) into the key elements of successful organisational change models [30,31,32,33], which included establishing motivation, identifying sponsors and leadership, building support for change, managing change transition, and sustaining momentum. The goal of this activity was to identify repetition and synergy across the information and identify evidence-based core components that each related to a key focus of action that could be standardised across settings, thus establishing the core components of uptake to be articulated in the SUSI framework.Grouping those core components into sequential phases to reflect both the learning from the PAR study’s real-time change transition and the steps that are widely used to implement existing organisational change models. The SUSI framework was then circulated for feedback amongst clinical and research experts through ten iterations. Feedback was requested regarding whether the core components were sufficient and easy to understand and whether the sequence for implementation was pragmatic and feasible. The responses were provided by email and at monthly meetings, reviewed by the Research Advisory Group, and the final analysis was conducted by two reviewers (CF and AS). The comments were analysed deductively using the key principles identified in the research evidence [40,41,42,43], and they were analysed inductively to identify any ideas expressed that were not identified in the literature. Disagreements were discussed by CF and AS, and the framework was recirculated to the respondents until a consensus was reached on the framework that accurately reflected the feedback. The goal of this activity was to organise the agreed core components into a sequence of steps that would have most efficiently and effectively resulted in the timelier uptake of IC into routine practice, thus establishing a pragmatic and feasible process for applying the SUSI framework in practice.

## 3. Results

### The Core Components of the SUSI Framework and Their Operationalisation

Figure 1 delineates the SUSI framework and identifies the order in which its core components should be delivered to optimise the efficiency of the uptake process. The six core components for the sustained uptake of the service innovations that were identified by combining the existing research evidence with the views of research and clinical experts are (1) forming an alliance with strategically identified stakeholders and securing commitment and active participation; (2) describing the current state of play; (3) building collaboration and a shared vision; (4) co-designing a service innovation [e.g., a tailored model of care]; (5) developing training and clinical support to deliver new procedures; and (6) building staff confidence and monitoring the uptake and outcomes. 

Table 2 identifies the change-enabling features of the core components, that is, the key reasons why each core component should be effective in achieving the uptake of a service delivery innovation. Table 2 also presents examples of the activities devised by staff in the PAR study to operationalise each core component. These activities were determined to be appropriate by the participants in the specific context of the uptake of IC in their MH and AOD services but are identified as flexible activities because these would need to be developed by stakeholders depending on the nature of the innovation they are implementing and the specific circumstances of their service delivery settings. The six core components remain the same across settings, however, and are described in detail after Table 2.

The six core components are described as follows:Core component 1. Secure Commitment and Active Participation.

This component is a critical first step. The goal is to identify relevant stakeholders and form an alliance by establishing commitment and active participation at multiple organisational levels. Successful engagement at different levels helps to ensure the feasible implementation of a substantial change to clinical service delivery. In addition to collaboration with non-government and community organisations, for example, a change in government-funded health service requires involvement from:*Service managers and clinical leads*. Clinicians are unlikely to engage in collaboration and practice change if their line supervisor is not actively supportive of the process. Clinical leads and service managers can ensure effective service planning and delivery, and they are well-positioned to lead and/or support innovations with staff and consumers. Participation at this level establishes IC as a priority and demonstrates to the staff that their involvement is valued and taken seriously. Involvement is also required from executive (or senior) managers, however, to ensure the clinical managers’ role expectations are met and their decisions are supported, for example, by allocating time for staff to take part in co-designed activities. If securing commitment and participation from clinical/service managers is successful, the process of securing a commitment from senior managers can then be taken appropriately and optimised for success.*Senior (Executive) managers*. Sustainable change requires a commitment to system-level change. For example, allocating designated leaders to drive the shift to IC increased the services’ capacity to maintain momentum in the change process whilst also allowing time to develop sustainability mechanisms, such as strategically identified Performance Indicators. Without input from the executive level about how to create and support designated leaders on the frontline, however, known barriers (e.g., competing clinical priorities, staff shortages, and organisational changes) are likely to prevent a successful change transition. This step acknowledges and draws upon expertise from the executive level, creates opportunities for access to other initiatives and resources, and ensures the proposed innovation aligns with the health services’ priorities and limitations.*Clinicians and frontline staff*. This step establishes staff confidence and interest in delivering IC and works to secure active participation from clinicians to drive the co-design and co-implementation process. Once participation is established, the implementation can draw on the clinical expertise that already exists within the workforce to improve staff confidence and capability. This step enables the development of practices that staff are more likely to adopt.*Clients of the participating health service (consumers, people with lived experience)*. Consumer participation is key to designing care that meets the needs of the people most likely to seek that care. Navigating complex systems can be difficult, but involvement from consumers from the design phase onward can help services to identify those difficulties as well as strategies to reduce their occurrence or impact. This step enables the development of practices that are acceptable and relevant to consumers and are, therefore, more likely to be used.
Core component 2. Describe the Current State of Play.

This second step helps to ensure that the proposed change is targeted appropriately and that the progress and impact can be monitored. For IC, for example, this includes determining the size of the problem (i.e., how many clients are likely to benefit from IC) and the levels of motivation toward providing IC (i.e., how do you define it, what does it look like in different services, and are you interested in/willing to try it?) and the strategic priorities of the organisation (does this innovation target a known area of need and/or respond to an established performance indicator), and conducting an assessment of the service capability and staff confidence and attitudes towards delivering IC. 

Core component 3. Formalise Collaboration and Build Relationships.

In this critical step, teams decide how and where they will co-design practices, e.g., for inclusion in the tailored model of care. Priority is placed on developing the collaborative partnership, which means allowing sufficient time to build genuine professional relationships between staff and between staff and consumers and encouraging a shared goal to emerge rather than be imposed. Consistent with the iterative, experiential learning approach of PAR, this component may overlap with core components 4 (co-design of the model) and 5 (develop training and clinical supports). 

Core component 4. Co-design the Innovation.

This step involves reviewing the existing practices and designing, trialling, and adapting new practices, e.g., for inclusion in a new model of care. Encouraging staff to play an active role in determining how new practices are structured, communicated to clients, and implemented facilitated meaningful contributions to the development of clinical procedures. Creating opportunities for clinicians to devise their own activities and methods fosters a sense of ownership over new procedures and a shared responsibility for ensuring uptake. In the specific context of IC, in which the SUSI was developed, the co-designed model of care is presented elsewhere and provides a detailed example of how core component 4 might be actioned [47]. This component may overlap with core component 3 (relationships) and core component 5 (developing clinical supports). 

Core component 5. Develop Training & Clinical Support (for delivering new procedures).

Ensuring that new practices are acceptable and feasible for clinicians to implement is critical to the uptake of the practices. This applies not only to those clinicians who actively participated in the co-design process but also to those within the participating services who did not play an active role. Increasing confidence and interest in delivering IC and in collaborating with other teams is key to improving staff satisfaction and shifting historical perceptions about the divide within or between services. Ensuring the appropriate clinical supervision for staff, both during and beyond the practice change, helps to foster staff satisfaction and sustainability. This component may overlap with core component 3 (relationships) and core component 4 (co-design). 

Core component 6. Monitor and Sustain the Innovation.

This step helps services operationalise the ‘refreezing’ or ‘sustaining momentum’ phase of change models. The goal is to embed sustainability mechanisms and monitor uptake of the innovation, and to build confidence among staff to deliver new procedures. Essential to this phase are (a) sustaining connections between staff; (b) maintaining a visible, collaborative presence; (c) embedding routine evaluation; and (d) choosing evaluation methods to monitor both process and outcome levels through key performance indicators (e.g., staff satisfaction and uptake of IC activities plus patient outcomes and experiences). A critical aim of this step is to transition short-term implementation and evaluation tasks into strategically identified embedded roles to support the sustained uptake of IC beyond the implementation phase.

## 4. Discussion

This paper presents the SUSI framework for guiding the process of implementing the sustained uptake of health service innovations using the example of moving from the independent delivery of AOD and MH services to IC. The SUSI comprises six core components, and these components need to be operationalised by individual services through a range of flexible activities. A clear pathway is established for actioning the core components in a sequence recommended to optimise the efficiency with which the service innovations are implemented: identifying the need to engage clinical managers first and then senior managers, clinicians, and consumers; taking the time to develop relationships between teams and allow a shared goal to emerge; co-designing an innovation (e.g., a model of care); developing training and clinical supports to support uptake of the innovation, and monitoring sustainability whilst building staff confidence to deliver new procedures. 

Although organisational change models exist [30,31,32,33], including Lewin’s 3-Stage Model [30] and Kotter’s 8-Step Model [31], they are heuristic guides rather than pragmatic frameworks that can be used to operationalise the specific activities that are required to give effect to the real-world implementation of innovations in service delivery across multiple and diverse locations. The SUSI framework, conversely, is a detailed and measurable blueprint for implementing a process of change that relies on the expertise of service providers to operationalise its six core components. An advantage of the SUSI framework is that it separates out levels of management in more detail than existing IC models. For example, it highlights the importance of engaging service managers and clinical leads to assist with the process of engaging executive managers because clinical and service managers are well-positioned to understand the day-to-day function, risk, and potential for change in frontline services. 

The core components of SUSI are standardised across any setting and are, therefore, comparable, and measurable, but their implementation needs to be co-designed by service provider experts to their own specific circumstances. In the study case of IC, for example, core component 1 (securing commitment and active participation at multiple organisational levels) was operationalised through identifying a manager as an advocate, presenting at existing meetings to secure support, seeking candidature in the Health Education Training Institute’s Rural Research Capacity Building Program, and holding regular collaborative meetings. Core component 6 (monitoring and sustaining the innovation) was operationalised by identifying the performance measures and mechanisms already in place for the routine monitoring of the outcomes. A key focus in the design of the SUSI framework was to ensure that it could be applied to a range of different health services and tailored to the specific characteristics of those services, which includes the use of established and meaningful measures. 

While the primary feature of the SUSI framework is its clear focus on being pragmatic, its key limitation is that it has not yet demonstrated its effectiveness in achieving timely service improvements in a range of different settings. Consequently, it clearly needs to be tested in prospective and well-controlled evaluation studies. Nevertheless, the SUSI framework is likely to be effective because its construction was based on the established principles of evidence-based practice [36,37,38,39] and the co-design of models of care, which are both standardised by the best evidence and operationalised by service provider experts [40,41,42,43,44]. Importantly, it also emerged from the real-world adoption of IC between AOD and MH services [34]. Unlike change models that utilise bottom-up or top-down approaches, the SUSI provides pragmatic guidance for change that is driven from the bottom, or frontline, up (to foster ownership amongst those responsible for enacting the change) and is visibly supported from the top-down (to enable permission and adequate resourcing). Testing and refinement of the SUSI are underway to ensure that it has the best possible chance of streamlining the application of effective innovations that are meaningful to communities and feasible for clinicians to implement in real-world settings. 

## 5. Conclusions

The SUSI framework was developed to respond to the complexity of integrating service innovations into the real-world delivery of health and human services. It complements existing organisational change models by providing a more pragmatic, step-by-step guide for the implementation of service delivery innovations in real-time. It also recognises and responds to the emerging idea that the translation of research evidence into practice is much more complex than rolling out evidence in clinical settings in a linear way. Rather, feasible implementation requires the tailoring of that evidence to fit specific services and settings without compromising the effectiveness of the innovation. A framework for the sustained uptake of service innovations is more likely to facilitate the embedding of research evidence efficiently and effectively into clinical practice if it is driven by frontline staff with leadership from consumers, and permission and support from middle and senior managers. The SUSI is underpinned by this leadership from those who are most affected by changes to healthcare services, that is, the people who receive health services, and those who deliver health services. Moreover, the SUSI explicitly addresses the need to tailor the best available research evidence to the routine delivery of services.

## Figures and Tables

**Figure 1 healthcare-11-01786-f001:**
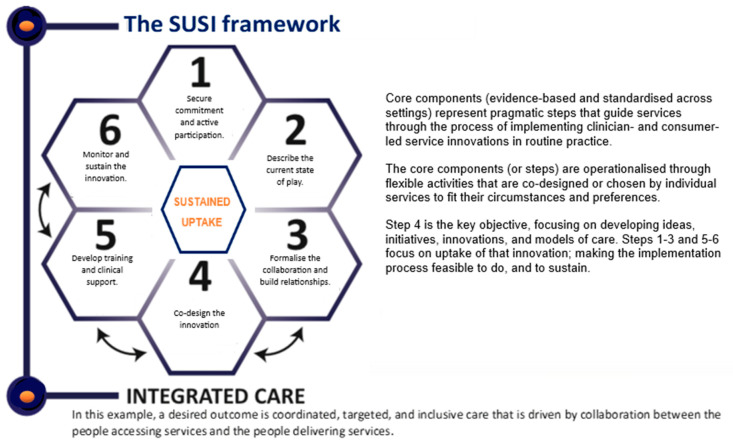
A pragmatic framework for the Sustained Uptake of Service Innovations (SUSI).

**Table 1 healthcare-11-01786-t001:** Procedure for adapting evidence-based practice into the SUSI framework.

Step	Description	Activity Undertaken
**1**	ASK a searchable question	Identified by clinicians and brought to researchers. Development of a PAR study. Question: How do you make IC work in practice across diverse clinical settings?
**2**	ACQUIRE information	Reviewed the literature [45] and conducted the PAR study [34]
**3**	APPRAISE the search results	(a) Reviewed clinical guidelines to ensure alignment with health services. (b) Triangulated findings from the PAR study, literature, guidelines (see Appendix A). (c) Drafted a framework (core components of uptake) via established process [40,41,42,43,44].
**4**	APPLY the evidence in practice. - For groups: develop a protocol and a change plan. - For individuals: document in the patient’s file.	(a) Sought consensus via element testing/feedback from academic and clinical groups. (b) Sought suggestions/agreement on names for the uptake framework. (c) Commenced development of studies to test the SUSI in multiple settings.
**5**	ASSESS the provided care	Results of testing will be published separately.

Note. Based on the procedure for adapting evidence-based practice for individuals or groups [37,38,39].

**Table 2 healthcare-11-01786-t002:** The core components, and flexible activities to operationalise them, of a pragmatic framework for the uptake of health service innovations.

Core Components	Change Enablers	Flexible Activities (Examples)
**1. Secure commitment and active participation.** Identify relevant stakeholders at all levels of the organisation and form an alliance, starting with service managers and clinical leads.	The innovation is actively driven by frontline managers, and workers (bottom-up leadership) alongside executive sponsors and managers (top-down permission and participation).	Research the idea, and relevant strategic priorities. Identify an advocate at service management level. Look for existing meetings to discuss ideas.
**2. Describe the current state.** Understand existing processes, strengths, and gaps. Establish a baseline.	The innovation is monitored for acceptability and effectiveness from commencement and ongoing, by the people delivering and/or receiving clinical care.	Ask your advocate or service manager about appropriate pathways, people, existing processes. Seek internal research support (e.g., the Health Education Training Institute’s Rural Research Capacity Building Program) and form a research advisory group.
**3. Formalise collaboration, build relationships.** Establish collaboration processes and a shared vision.	There is sufficient time for genuine professional relationships to develop, and a shared goal to emerge rather than be imposed.	Form a group of interested clinicians to lead. Hold regular, purposeful get-togethers that are open, and welcoming, and visible to staff and clients. Connect via site visits, shadowing, shared events.
**4. Co-design the innovation.** E.g., design a new model of IC or tailor an existing model to the needs of your service.	Buy-in and ownership are optimised through equitable & meaningful contribution to development of clinical procedures, and opportunities to devise own activities.	List existing practices and processes. Develop shared activities, trial, & improve. Tap into staff creativity, draw on client expertise.
**5. Develop training and clinical support.** Co-design training and processes to support clinicians to deliver new procedures.	Training and clinical supports are relevant and productive and are delivered in a flexible and feasible way.	Review staff needs and current processes that could be enhanced or optimised. Identify opportunities for peer support, skill sharing.
**6. Monitor and sustain the innovation.** Embed measures and continue to build staff confidence via clinical supervision and supports.	The innovation is embedded in routine practice and clinicians have confidence to deliver the new procedures.	Build strategically identified performance indicators into identified positions to monitor workforce, treatment, and systemic activities.

## Data Availability

The data presented in this study are available upon reasonable request from the corresponding author. The data are not publicly available due to HREC requirements for confidentiality and to prevent the participants from being identified.

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
