# Peer review of "Utilising the Implementation of Integrated Care to Develop a Pragmatic Framework for the Sustained Uptake of Service Innovations (SUSI)"

_healthcare, 2023, doi:10.3390/healthcare11121786_

Round 1
Reviewer 1 Report
Thank you for allowing me to review the manuscript.Integrated care is recommended in many countries. The benefits are avoided alcohol relapse, reduce mental deterioration and reduces the cost of treatments.
The introduction is difficult to read. The tables are to busy.
In general, the text is difficult to understand; which makes the manuscript a bit confusing
Author Response
Dear Reviewer, thank you for taking the time to review our paper and to provide useful feedback. We have incorporated your suggestions and the manuscript is substantially improved. Warm regards.
Point 1. The introduction is difficult to read.
Response 1. Revised and streamlined with attention to flow and readability.
Point 2. The tables are too busy.
Response 2. Revised and streamlined.
- Table 1 moved to Appendices.
- Text in Table 2 (now Table 1), in Table 3 (now Table 2) and in Figure 1 was reduced. Figure 1 is now presented in a different format.
Point 3. In general, the text is difficult to understand which makes the manuscript a bit confusing.
Response 3. Revised and streamlined throughout with attention to flow and readability. Attention to methods section.
Reviewer 2 Report
Thank you for sending me this paper to review. It is an excellent piece of work which will positively contribute to the literature. I am not an expert on IC but it is well described and evaluated using relevant citations. I have however written and taught on various theoretical/conceptual models and agree that one of the common limitations is that they are not presented or developed in a manner that is pragmatic and easily interpretable by clinicians. This paper addresses that gap and can be used by other researchers who are developing conceptual models/frameworks.
On a minor note, I mentioned in my feedback that the language required minor correction. That was simply because no option was provided in the review form to indicate the language did not require improvement.
Well done.
Author Response
Dear Reviewer, thank you for taking the time to review our paper and to provide comprehensive and feedback. We appreciate your comments, and we will use them to continue to improve our work. We have responded to other reviewers’ suggestions about streamlining the introduction, methods and tables and we hope that the manuscript is improved. Warm regards.
Reviewer 3 Report
Dear Authors.
I would like to congratulate you on the manuscript you have prepared.
I consider it important that the conclusion of the study contains current recommendations, as well as being better articulated with the results and the discussion of the results.
Best regards and keep up the good work.
Author Response
Dear Reviewer, thank you for taking the time to review our paper and to provide useful feedback. We have responded to your suggestions and the manuscript is substantially improved. Warm regards.
Point 1. I consider it important that the conclusion of the study contains current recommendations, as well as being better articulated with the results and the discussion of the results.
Response 1. Revised and streamlined with attention to providing clear recommendations and improved discussion.